# Building a Special Representation for the Chinese Ancient Buildings in Diffusion models

## Abstract

Benefit from the great generative ability of diffusion models, people can build various images based on their imaginations via some carefully designing prompts. Acctually, the functional blocks, like CLIP, for the alignment between prompts and representation of images plays the key role. Limited by the training data, these models performs worse in some rare areas, like Chinese ancient buildings. The reason comes from the missing of special representation of these building's elements, such as breckets, roofs, bias of different periods. In this paper, we firstly collect more than 400 images of ancient buildings. Several subsets are separated by their generalities. Secondly, pinyin, the basic tool for learning Chinese, is firstly introduced into large models as the specific tools to describe the characters of these buildings. Thirdly, we train several fine-tuning models to exhibit the ideal performance of our models compared with existing models. Experiments prove that our frame can resolve the barriers between English-centric models and other cultures.

## 1 Introduction

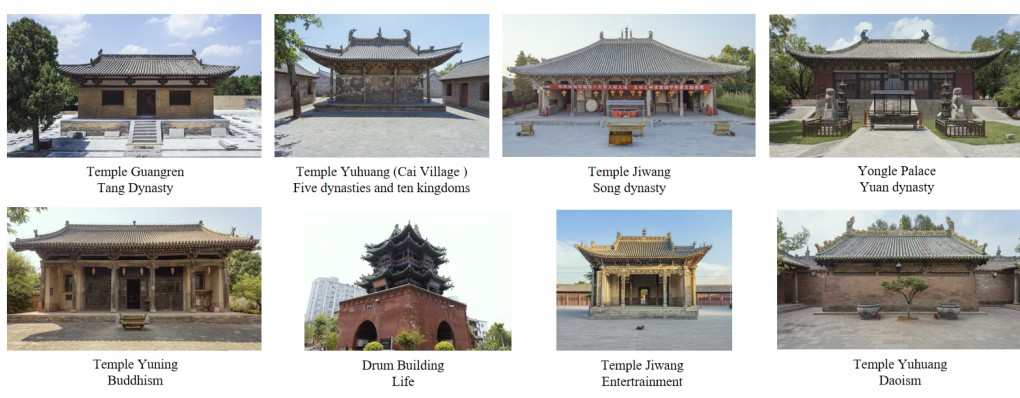

Figure 1: Some typical samples of the Chinese ancient buildings. To show the difference, some key figures are listed below these figures. On the top, these buildings are built from Tang dynasty to Yuan dynasty, while the below ones are classified by their function. There are signiture difference from roof to the shape.

In the past two years, our life is completely changed by the tremendous break both in NLP and CV area. Big models in both of CV and NLP area, like ChatGPT and Stable Diffsion, are the most frequent topic in not only AI research circles, but also for normal people. These models open the door to AGI world. Various satisfied results can be obtained by different prompts. There are some reasons for this huge developments. The basic one comes from big data that Internet users all over the world provides. As mentioned in the (GPT), the ,,, provide the basic languages for training. In CV area, as said in the paper ..., the outstanding performance comes from the professional image dataset....Besides, developments of deep learning models in text-to-image contributes to the boosting improvements. Researchers develop creative architectures and theories for these generative models.

Despite existing larger models performs satisfied in well-studied scene. In specified area, training or fine-tuning strategy are still challenging, as it requires both domain knowledge and ML expertise Shen et al. (2023). It can alleviate the potentially challenge by applying fine-tuning workflow with fixed dataset. Recent works have demonstrated the possibility of fine-tuning pre-trained models to other tasks, like vision tasks Dinh et al. (2022); Lu et al. (2023), NLP area Bakker et al. (2022); Hu et al. (2023), and reinforcement learning Reid et al. (2022). A common sense can be obtained from these approaches that fine-tuning format can address the issue between generality and specific-task in cross-domain learning.

In this paper, we focus on the generating of the Chinese ancient building area. Noted that the Chinese ancient buildings vary a lot not only for different period, but also for different grade. As shown in Fig. 1, the buildings in different dynasty carry some unique charateristics. All these bring great challenge in the generative models. If we want to obtain some similar ones, carefully designed prompts and reasonable features must be learned during training. Mostly, satisfied images is rarely generated by Midjourney or SD. There are two important reasons: The first one comes from the fact that most diffusion models are trained following the English culture, and the specific purpose in other culture or language have completely logistic structure compared with English. The second reason is that English words are sufficient for the expression of western culture and exhibit poor expression ability in other language situations. For other area, there are a lot of unique definition in the Chinese ancient buildings, like various roof. As a result, the personalization text-to-image models raise as a challenge in some specific task.

To overcome the issue in the Chinese ancient building, we generally provide a framework from data-collecting to prompt designing. The contributions is as follows:

- We take hundreds of professional photos of ancient buildings. To cover details as much as possible, a lot of views are included, like front, back, top, side, covered-lens, long-lens views, and so on. To the best of our knowledge, this is the first trial about the ancient buildings built in Yuan dynasty and pre-Yuan dynasty.
- Based on limited computing power and efficient framework, two fine-tuning strategies, Lora and Prompt, are introduced in our experiments and result in satisfied results with carefully designed prompts.
- We introduce the Pinyin mechanism into the prompts project for few-shot personalization. In order to closer to the description and definition in Chinese, it is regarded as the identified solutions for the vivid requirements in the Chinese ancient building area.
- To evaluate the new task, we design a lot of experiments to proof the outstanding performance in Chinese ancient building generating area. The generated results exhibits subject fidelity and prompt fidelity according to the data characteristics and pinyin settings.

## 2 RELATED WORKS

### 2.1 DIFFUSION FOR MULTI-MODELS (CONTROLLABLE GENERATIVE MODELS)

Generative Models, to produce high-quality and vivid images, has received increasing attention in recent years. In computer vision area, traditional methods, like generative adversarial network(GAN), variational autoencoders(VAE), Normalizing Flows, assume the hidden distributions to represent varies features of all images, which aims to yield realistic images in an image-to-image wayGoodfellow et al. (2014)Chen et al. (2020)Karras et al. (2019)Kingma & Welling (2013)Bao et al. (2017)Huang et al. (2018)Dinh et al. (2014)Kingma & Dhariwal (2018). However, conventional generative models tend to falter when faced with detailed controllability and personalization requirements. Inspired by these early methods, diffusion probabilistic model was proposed to learn training data distribution in denoising processHo et al. (2020)Song et al. (2022). Other effective techniqueDhariwal & Nichol (2021) was proposed to use a classifier to guide a diffusion model during sampling ,and found that it surpass state-of-the-art GANs model. PaletteSaharia et al. (2022a) realized image-to-image transaction based on conditional diffusion model. Despite the emergence of diffusion model for image-to-image generation, multimodal method are under explored. For latent diffusion modelsRombach et al. (2022), like stable diffusion, the joint of pretrained language model Contrastive Language-Image Pretraining (CLIP)Radford et al. (2021) arise the viewed definition of hidden information by words. Other effective techniques such as DALL-E2Ramesh et al.

(2022) and ImagenSaharia et al. (2022b) made remarkable progresses in TTI. Moreover, new approach SDXLPodell et al. (2023) designed a improved stable diffusion framework. Despite their great performance, these method are not compatible with other language.

## 2.2 FINE-TUNING METHODS

In general, most available models work as a general tool which mean various features are transferred and learnt by one model. However, as a common sense, general will damage the representing ability in specific area, like Chinese traditional building area. The fine-tuning machinist can be an efficient tool to ease the problem. Fine-tuning machinist originally appeared in field of natural language process (NLP)( Devlin et al. (2018);Howard & Ruder (2018)), due to the great performance in application, the machinist was introduced into computer vision generative models (Gal et al. (2022);Ruiz et al. (2023);Kumari et al. (2023);Zhang & Agrawala (2023)). Fine-tuning machinist is a approach that fine-tuning the pre-trained parameters to downstream tasks. Originally, fine-tuning machinist named full fine-tuning simply retrain all pre-trained parameters, however challenging computing power and memory ( Radford et al. (2018)).

**Parameter-efficient Fine-tuning** To save training costs, Parameter-efficient fine-tuning only retrain a smaller set derived from pre-trained parameter set. There are a number of measures to reduce parameters. Some researchers focus on training a distillation model to generate a simple student model to save costs (Li et al. (2021);Wei et al. (2022)). Further more, Cai et al. (2019) propose a Progressive Shrinking algorithm to distill smaller sub-network progressively and greatly improving the training efficiency. Adapter training machinist insert adapter modules between layers of initial model and only fine-tune adapter modules to save memory (Houlsby et al. (2019);Lin et al. (2020)). However, adapter module is still complex in application. Hu et al. (2021) propose LoRA structure which introduce rank decomposition matrices into each layer , greatly reducing the number of trainable parameters in complex task. LoRA is the main method used in our paper. At last, some researchers focus on input word embeddings instead of agent model(Li & Liang (2021);Lester et al. (2021)).

In this papar, we focus on the fine-tuning strategies of a pre-trained text-to-image diffusion model. A creative object and new data are organized and several fine-tuning strategies are introduced to explore the representation ability.

## 3 METHODS

Mostly, the fine-tuning process includes pre-trained models, fine-tuning mechanism, data-editing. For specific area, there are a lot of preliminaries to modify. Inevitable conflicts will raise as a big challenge, especially for the ancient buildings across English and Chinese. We firstly provide some background of the referred diffusion models and fine-tuning modules in this paper(Sec. 3.1). Secondly, we present the preliminaries which we faced during implying these model in our research (Sec. 3.2). Finally, we propose the specifics of our innovation to overcome the challenge in Chinese ancient building genetation task (Sec. 3.3).

## 3.1 DIFFUSION MODELS AND FINE-TUNING

In this paper,the diffusion model is used as the basis for its strong image prior. As a generative model, diffusion models are trained to learnt the data distribution $P(x)$ by gradually denoising the sampled variable $z$ which follows the Guassian distribution$p(z)$. With the trained model, we can generate various samples with intended will.

Specifically, our research leverage a popular text-to-image generation model, Stable Diffusion (SD) Rombach et al. (2022), in consideration of two main reasons. The model is trained with a lot of high-quality images including various classes. As a result, plausible image can be obtained from the strong generating capability. Secondly, the CLIP Radford et al. (2021) model is utilized to successfully implement Zero-shot learning ability with some chosen language information. Moreover, the CLIP realizes the representation of embedded images and text, making the text-to-image task with a good initialization. Compared with prior CV models, which learn the latent features from processed inputs, the text-to-image diffusion models essentially focus on understanding the align-

ment between feature and text. As a result, satisfactory images can be generated by prompts as large-language models(LLM) in NLP area. In this paper, We leverage a pre-trained SD similiar as in Ruiz et al. (2023); Kim et al. (2022).

Started from the input images $X_0$, the noise $\epsilon$, following the standard normal distribution $N(0, 1)$ is gradually added with several steps $t$, $t = 1, \cdots, T$. In each step, the process like $q(x_t|x_{t-1}) := P(\sqrt{1 - \beta_t}X_{t-1}, \beta_t \boldsymbol{I})$, where the $\beta_t, t = 1, \cdots, T$ is a hyper-parameter or learned variance. The final latent $x_T$ can be writen as:

$$X_t = \sqrt{\alpha_t}x_0 + (1 - \alpha_t)\omega, \omega \sim P(0, 1)$$

where the $\alpha_t := \prod_{i=1}^{t}(1 - \beta_i)$. This set of assumptions describes a stable and gradual noisification progress of the input image $x_0$. There is no learned parameter $\theta$ since this forward is modeled with fixed or learnt parameters, including mean and variance. As a result, the reverse process $P_\theta(x_{t-1}|x_t)$ plays a crucial role if we want to simulate a new data. General speaking, the Gaussian noise sampled $P(x_T)$ combines with a conditioning vector $c = \tau(P)$, where $\tau$ refers to a text encoder and P is the prompt, as the image and text embedding. The combined embedding will iteratively denoise with the transitions $P_\theta(x_{t-1}|x_t)$. A ideal image $x_0$ with specific prompts will generate after $T$ steps. The parameters $\theta$ can be learnt by addressing the optimation function:

$$L = \mathbb{E}_{x,c,\epsilon,t}[w_t||\hat{x}_\theta(\alpha_t x + \sigma_t \epsilon, c) - x||_2^2]. \tag{1}$$

In this function, the x refers to the ground-truth, and c is the conditioning vector from the CLIP. Moreover, the $\alpha_t, \sigma_t, w_t$ is some parameters for the noise schedule and sampling control. Finally, the frame of our model is shown in Fig 2. It si similar to the CLIP-diffusion Kim et al. (2022) model.

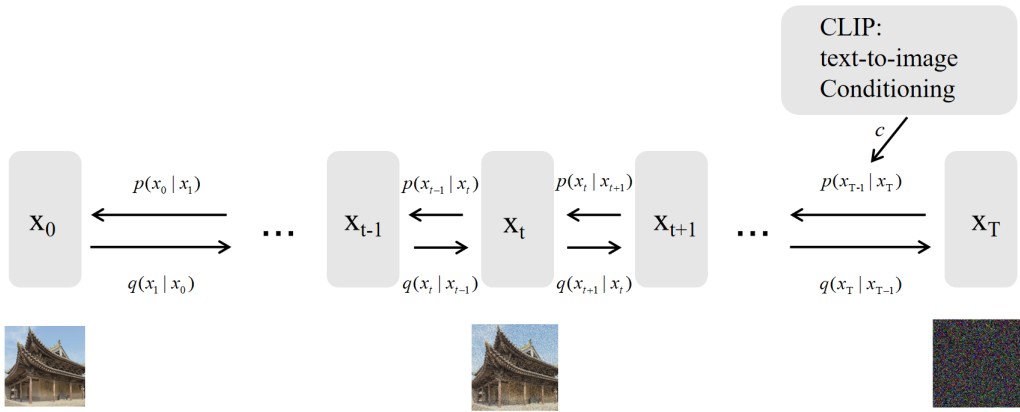

Figure 2: Overview of the text-to-image diffusion model. The input image is first converted to the latent as some specific Gaussian distribution via the diffusion model. In the reverse process, combined with the prompts, the diffusion model can denoise from latent space with fixed direction, and the updated sample is progressively generated during reverse steps.

In this paper, we combine two fine-tuning methods to exam the performance of our setting. Intuitively ,the first fine-tuning method is prompt optimization. Secondly, the Lora module is adopted to train a class-specific model. The reasons behind these fine-tuning operations are listed in the following parts.

## 3.2 PRELIMINARIES

Nowadays, most diffusion models can be viewed as general models which means that their performance in a particular area can have a big decline. The importance of high-quality dataset arise when these pre-trained models apply in specific area. Besides, the problem of overfitting or model-collapse should be considered during the training. For text-to-prompt models, researchers should highlight the importance of languages' expressive skills. The key words of the introduced object must describe unique features and easy to be understood.

To address these issues, the first task is to gather enough subject instance into the unseen domain. Hence, the pretrained diffusion model can learn the novel representation of the new subject. Moreover, the quality and overlay perspective should be as comprehensive as possible. A multi-hierarchical dataset which includes several subcategories of the object raise to be an effective solution. And then, a mix LORA model or several sub-LORA models increase research space for solutions.

Other challenge is how to design suitable prompts for specific area. For multi modalities, text and images in our task ,we can think of the data we observe as represented or generated by an associated unseen latent variable, which we can denote by random variable $z$. Both modalities should match each other in the $Z$ space. It means that the expression of words is consistent with the representation of images. However, the expression ability of existing tokens for rare conditions shows insufficient performance. Designing precise prompts of the detailed image descriptions for a given data set plays the central role. Moreover, these prompts should be simple and featured. Inflenced by the Ruiz et al. (2023), we choose the Pinyin as the key solution of the Chinese language. As far as know, the pinyin mechanism have powerful expression ability and it is used for the Chinese beginners. Unlike the mentioned method in Ruiz et al. (2023), the pinyin system inherits the logistic thinking of Chinese defines and expresses. It involve no extra letter beside the English alphabet. As a result, the introducing of pinyin mechanism in text-to-image models is not only easy to understand, but also act as a rare identifier.

### 3.3 SPECIFICS OF THE TRADITIONAL CHINESE BUILDINGS

Referred to the Chinese ancient building, there are a lot of unique characteristics, like the roof, brackets, ceiling, appellation, and so on. Take the roof as an example, there are thirteen main categories. We list six different classes in the Fig. 3. Unlike the western buildings, the political hierarchy and protection have the closest influence on the design and architecture of the Chinese ancient buildings. Nowadays, most diffusion models are trained on a massive amount of English data. As a result, the diffusion models is not trivial to sample satisfied results in Chinese. One important reason attributed to the missing captions of another language in the CLIP.

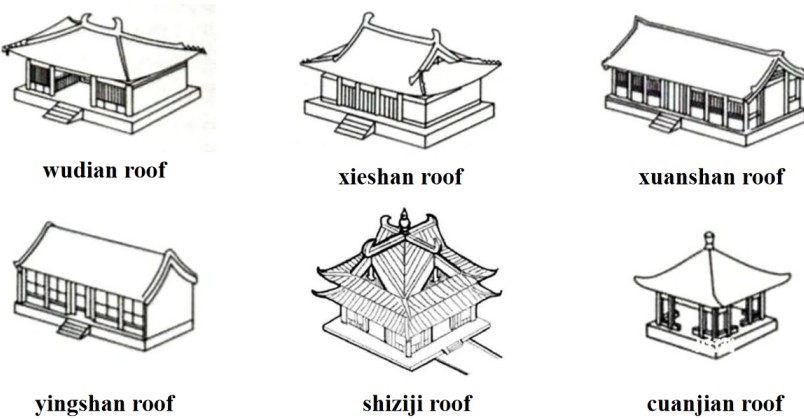

Figure 3: Six examples of the roof related to the Chinese ancient building. Different from the western buildings, which always defined by the shape, Chinese roof is more influenced by political hierarchy. Like the xuanshan and yingshan roof, the yingshan roof is about fireproof and the xuanshan relates to the watertight.

Design corresponding prompts for personalization. The words of Pinyin for Specific Language to identify the Chinese buildings. By simply combining the noise predicted from several per-trained lora models, multiple attributes can be changed simultaneously through only one sampling process.

## 4 EXPERIMENTS

### 4.1 SETTINGS

**Experimental settings.** Our framework is based on a popular text-to-image generation model, Stable Diffusion (SD) Rombach et al. (2022). The LoRA training was implemented using Kohya_ss bmaltais (2023). Specifically, we change different setting to evaluated our method: (1)We train the front view, side view, top view and back view LoRA separately or together. (2)We add pinyin during training process or just use complete English description. For detailed training, the learning rate is set as 0.0001 using cosine_with_restarts. We conduct training with a batch size of 2. We train 10 epochs and save a model every 2 epochs. The training time is 11 minutes every epoch(front view) on single GeForce RTX 4090 GPU.

### 4.2 DATASET

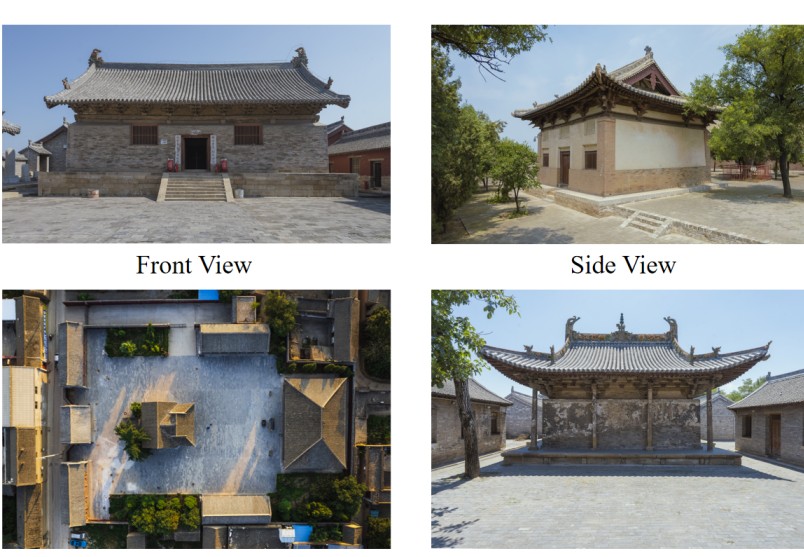

Front View        Side View

Top View        Back View

Figure 4: Datasets in different views. Four different views are chosen as the standard position when we take these photos. The front view is the most important one for these Chinese ancient builds, and the other views will supply additional details.

**Datasets.** As mentioned in the former parts, we take hundreds of images about the Chinese ancient buildings. The construction date of these buildings can be traced from Tang dynasty to Yuan dynasty. The time starts from 1200 years before and the time span is up to six hundred years. Some representative buildings are shown in the Fig 1. We will make the dataset public after the review of the paper.

Nowadays, there is no existing material about the image dataset of Chinese ancient buildings. The unique characteristics of Chinese traditional building, including the name of different parts, like roof, dougong, temple, period of building, and so on, still keep in an untapped state. On the other hand, most generative models use the images of the Forbidden city or some modified building to train their model.

To fill the gap, we collect more than 1200 images, and cover various Chinese ancient buildings like Buddhist temples, Taoist temples, Confucian temples, drum towers, local gods temples, etc. There are some key details. Firstly, all these images are toke by professional camera and resize to 3000*2000. As we know, the resolution means the amont of information in the images. The large size will provide more possibilities for follow-up tasks. In the experiment of this paper, we first set the image size as $768 \times 512$ which is suggested in the fine-tuning work of SD1.5 model. Secondly, we divide the whole dataset into 4 sub-datasets according to the difference of shooting perspective:

front view, side view, top view and back view. As a common sence, buildings vary a lot if we stay in different position. To provide information as much as possible, images must contain details from different views. In our different directions fine-tuning strategy, we select more than 30 images for each view in training. Some examples of different views are shown in 4.

Notably, due to the insufficient textual descriptions of images processed by stable diffusion, we add pinyin and detailed text prompts.

## 4.3 Comparisons

In this section, we assessed our method with various conditions. First, we evaluate the capacity of pinyin. We first compare the effect of whether the prompts are added with brackets(dougong) or not. The visualized results5 shows that our method can effectively captures the texture and structure of brackets. To better illustrate the effectiveness of pinyin, in particular, we compare whether pinyin is used in training and testing phase of LoRA in different perspectives in 5. When training LoRA without using pinyin and the prompts do not use pinyin, the building shows limitation in layout and structure in detail. Other conditions including LoRA without pinyin or prompts without pinyin have inaccurate results, which is due to the mismatch between the text and image. For a more detailed comparison, we fix the seed and compare our approach with Chinese ancient building without trained LoRA and modern buildings generated by SD in 6. We find that the generated ancient buildings from SD fail to capture the realistic style of Chinese buildings. And there is a huge gap between the modern buildings and ancient architectures.

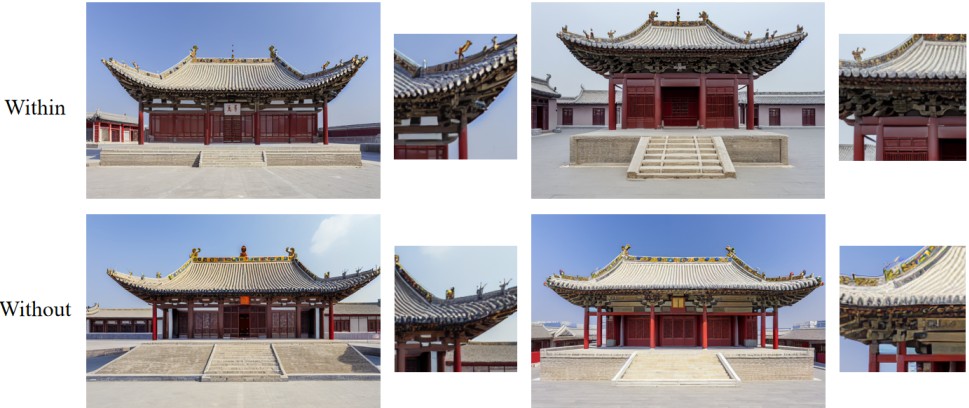

Positive Prompt: a chinese ancient building with xieshan roof, **dougong,** <lora:front_building:0.6>
<lora:side_building:0.4> best quality, masterpiece, 4k, front view, super detailed, symmetrical, clean background, cloudy
Negative prompt: worst quality, low quality, watermark, green, trees, human, colorful, shadow
Settings: Steps: 20, Sampler: DPM++ 2M Karras, CFG scale: 7, Seed: 310784348, Size: 768x512, Model hash: 6ce0161689, Model: v1-5-pruned-emaonly, Lora hashes: "front_building: a7d8a21eeb0c, side_building: 80f17f34ed74", Version: v1.4.1

Figure 5: Comparison of the effectiveness of pinyin, taking dougong(pinyin of bracket) as an example. The within row refers to the fact that the Pinyin words is used for generating. The without row remove the pinyin word. Noted that No more changes in the comparing experiment.

Then, we compare the different combinations of LoRA for the purpose of the optimal result. According the results, we observe that LoRA trained separately based on different perspectives (front view, side view and back view) is better than combined training. In order to find the optimal combination, we adjust the weights of the front, side and back LoRA. We find that our method trained with separately LORA models achieves best performance.

In order to further verify the effectiveness of our method, we implement our experiment using SDXLPodell et al. (2023). As shown in 7, we provide a comparison of our method and SDXL without LoRA. Specifically, SDXL can generate Chinese ancient style images with soft light, however, the results of SDXL lacks delicate textures and the unique characteristics of Chinese ancient

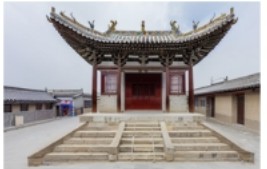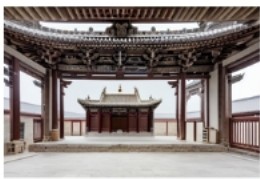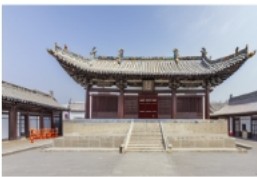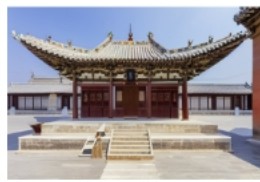

Prompt: a chinese ancient building with xieshan roof, dougong, <lora:front_building:0.6> <lora:side_building:0.4> best quality, masterpiece, 4k, front view, super detailed, symmetrical, clean background, cloudy
Negative prompt: worst quality, low quality, watermark, green, trees, human, colorful, shadow
Steps: 20, Sampler: DPM++ 2M Karras, CFG scale: 7, Seed: 1879481607, Size: 768x512, Model hash: 6ce0161689, **Model: v1-5-pruned-emaonly, Lora hashes: "front_building:** a7d8a21eeb0c, side_building: 80f17f34ed74", Version: v1.4.1

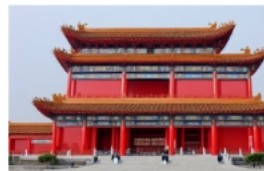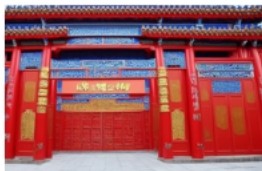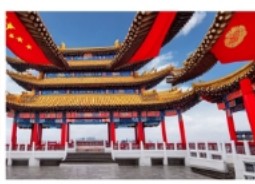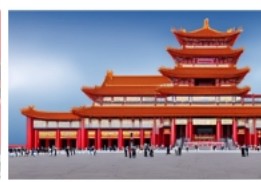

Prompt:a Chinese building, best quality, masterpiece, 4k, front view, super detailed, symmetrical, clean background, cloudy
Negative prompt: worst quality, low quality, watermark, green, trees, human, colorful, shadow
Steps: 20, Sampler: DPM++ 2M Karras, CFG scale: 7, Seed: 1879481607, Size: 768x512, Model hash: 6ce0161689, **Model: v1-5-pruned-emaonly,** Version: v1.4.1

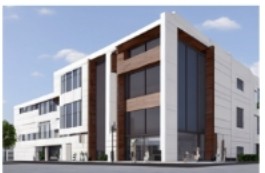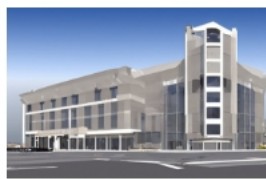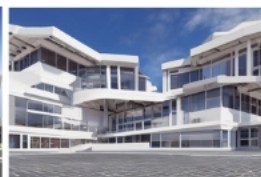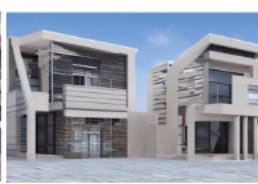

Prompt:a modern building, best quality, masterpiece, 4k, front view, super detailed, symmetrical, clean background
Negative prompt: worst quality, low quality, watermark, green, trees, human, colorful, shadow
Steps: 20, Sampler: DPM++ 2M Karras, CFG scale: 7, Seed: 1879481607, Size: 768x512, Model hash: 6ce0161689, **Model: v1-5-pruned-emaonly,** Version: v1.4.1

Figure 6: Qualitative results in Stable diffusion1.5, comparison with other approaches. To varify the useful of our proposed framework, we design three different settings during the text-to-image generation task. This first one includes some pinyin key words and two LORA models which are trained in our dataset. The second one remove LORA models and some key words, like ancient, dougong. The last one removes ancient to check the changes. Experiments prove the outstanding performance of our framework.

building. In contrast, our approach using SDXL with trained LoRA can generates vivid Chinese traditional buildings and better preserves the Chinese ancient architecture. For example, we can observed the results by our method retain the detailed architecture of roof like dougong in the second row. Meanwhile, the colors of our results are closer to ancient buildings and the image style is more realistic. The experiments can prove that using the proposed method can achieve outstanding performance in the Chinese ancient building area.

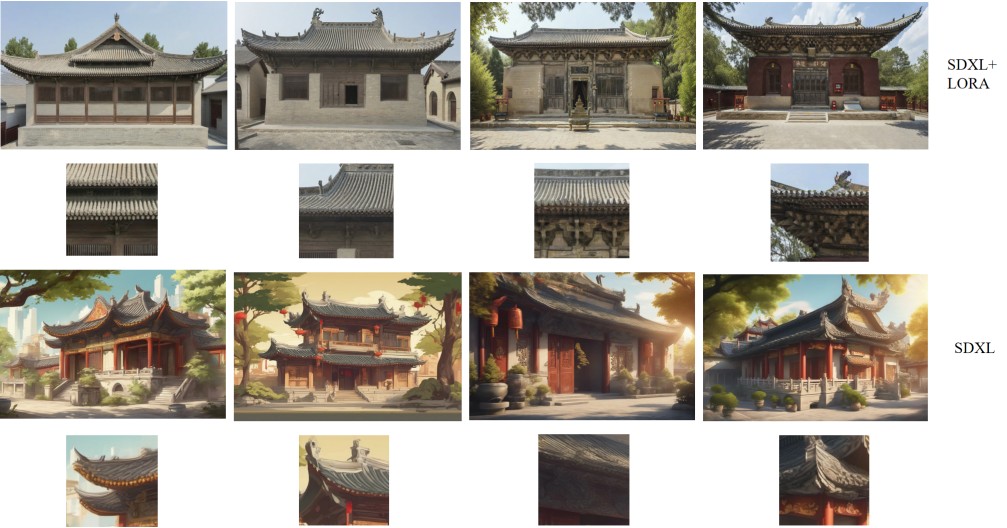

Figure 7: Comparison results with SDXL. We set the prompts as: Chinese ancient building, xieshan roof, roof dragon, a building with a censer on the front of it, one door, two windows, a sunny day, trees in the background. Some details are shown in the figure. We can see that our fine-tuning brings in realistic results compared with the original SDXL model.

### 4.4 LIMITATIONS

However, for the limitation of computing resource and time, more comparative experiments should be systematically done, like the evaluation indicators, combination with other tasks, and so on. We will keep on working in the Chinese culture generation tasks in future work.

## 5 CONCLUSION

With the outstanding performance of text-to-image diffusion models, how to imply these models in specific area raises to be a big challenge. In this paper, we focus on the Chinese ancient building generating task. It is a typical cross-cultural issue. To address it, we build a building dataset including four sub-datasets according to the shooting angle. Moreover, to overcome the language difference, we introduce the Pinyin mechanism into the fine-tuning strategy. Experiments prove the outstanding performance of our framework. In future, we will deeply analyze the characteristics of the dataset according to the peculiarity of the text-to-image models.

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
