# OpenReview forum: "Building a Special Representation for the Chinese Ancient Buildings in Diffusion models."
_ICLR.cc/2024/Conference — Submitted to ICLR 2024_

### Official Review · Reviewer_3LWK · 2023-10-25

**Soundness:** 1 poor
**Presentation:** 1 poor
**Contribution:** 1 poor
**Rating:** 1
**Confidence:** 4

**Summary:**

This paper propose to use diffusion models to generate images of Chinese ancient buildings. The authors adopt pinyin and LoRA to finetune the text encoder of the diffusion model. The experimental results show that the pinyin expression is better than the English expression and the LoRA is better than prompt tuning.

**Strengths:**

The experimental results show that pinyin and LoRA are effective.

**Weaknesses:**

1. Lack of novelty: the LoRA is an existing approach.
2. Lack of contribution: the usage of pinyin is trivial.

**Questions:**

What is the technical novelty of this paper?

---

### Official Review · Reviewer_Wuxm · 2023-10-26

**Soundness:** 3 good
**Presentation:** 3 good
**Contribution:** 2 fair
**Rating:** 5
**Confidence:** 4

**Summary:**

This paper aims to build representation for Chinese ancient buildings with diffusion models. The authors firstly collect the pictures of the buildings to form the dataset. Then they develop the representation using Pinyin sequence as the prompt input with fine-tuning and LoRA strategies on the diffusion model. Finally, they design experiments to proof the outstanding performance in Chinese ancient building generating area with the learned prompt. This can benefit some specific area like Chinese ancient-style building generation and some down-stream tasks in this community, if any.

**Strengths:**

The propose of this benchmark can benefit some specific area like Chinese ancient-style building generation and some down-stream tasks in this community, if any.

**Weaknesses:**

1.	The benchmark is novel and with great effort, while the learned representation and the prompt design are relatively simple. Though the diffusion model is not trained with such specific data, the pre-trained diffusion model still has the ability to generate realistic Chinese ancient buildings (with detailed caption keywords like ‘realistic’). The paper should provide such detailed comparisons.
2.	Lacking qualitative and quantitative comparisons with well-defined metrics like FIDs, CLIPIQA or so to validate the necessity of this benchmark and the special representation development.
3.	Why the Pinyin sequence? Is there a more effective prompt? The authors should provide a more detailed explanation.

**Questions:**

Why the Pinyin sequence? Is there a more effective prompt? The authors should provide a more detailed explanation.

---

### Official Review · Reviewer_szHe · 2023-10-31

**Soundness:** 2 fair
**Presentation:** 1 poor
**Contribution:** 2 fair
**Rating:** 3
**Confidence:** 2

**Summary:**

In this paper, the authors proposed to generate Chinese Ancient Buildings with Diffusion models, which is a very interesting  topic. The authors apply pre-trained diffusion models and fine tune it into the specific area where the lora is used. In addition, the authors proposed to use Pinyin as prompts in this specific topic. Furthermore, the authors collect one dataset of Chinese Ancient buildings with about 1200 large-resolution image, which will be benefited for the community. The experimental results by the visual comparison shows the proposed generation methods perform well.

**Strengths:**

I think there are several strengths attracting me:
(1) The topic is meaningful and interesting. I appreciate the authors' work on this topic, including the trial of applying novel technology (e.g., diffusion and lora) and the experimental results.
(2) The collected dataset seems useful for the related community.
(3) The authors introduced pinyin as prompts into diffusion models, which is promising.

**Weaknesses:**

(1) I think the big issue of this paper is about the paper-writing, even I suspect some places are not completed. For example, what does the symbols ",,," and "..." mean in the first paragraph?. Also, many writing typos and grammatical mistakes. For example, the sentence “It si similar to the CLIP-diffusion model” seems to be “It is similar to the CLIP-diffusion model”. I suggest that the authors proofread the paper carefully to avoid these.

(2) It is suggested to check the format and completeness of references in the paper. For example, it is incorrect to list reference as Shen et al. (2023), it should be (Shen et al., 2023). In addition, there are some places where references are missing, such as the sentence “Big models in both of CV and NLP area, like ChatGPT and Stable Diffusion” in INTRODUCTION.

(3) The way of citing pictures in the paper is suggested to use Fig. 5, 6 instead of directly an index.  For example, “the visualized results5”, “in different perspectives in 5” and “by SD in 6” in Section 4.3.

(4) In the experimental results, there are only some visualization results. It is suggested that the authors could add more quantitative comparison.

**Questions:**

Utilizing pinyin sounds reasonable and visualization results shows its effectiveness. However, what about other prompts?

---

### Meta-Review · Area_Chair_Ur5F · 2023-12-06

**Metareview:**

This paper introduces a novel application of diffusion models for generating images depicting Chinese ancient buildings, incorporating pinyin and LoRA for fine-tuning the text encoder within the diffusion model framework. The experimental findings assert the superiority of pinyin over English expression and highlight the efficacy of LoRA compared to prompt tuning.

However, a consensus emerges among reviewers regarding the paper's perceived lack of novelty. The utilization of LoRA is identified as an existing approach, diminishing the technical contribution of the proposed methodology. While some reviewers acknowledge the authors' commendable efforts in dataset creation and the introduction of pinyin prompts, the overall sentiment is that the paper falls short of providing a significant advancement in the field. Despite reviewers' appreciation for certain aspects, the absence of a rebuttal from the authors leaves the identified concerns unaddressed, contributing to the unanimous negative review scores.

**Justification For Why Not Higher Score:**

The authors did not provide a rebuttal and all review scores were not positive at the end.

**Justification For Why Not Lower Score:**

N/A

---

### Decision · Program_Chairs · 2024-01-16

Reject